# Analysis of Digital Teacher Education: Key Aspects for Bridging the Digital Divide and Improving the Teaching–Learning Process

Sonia Val [1,*] and Helena López-Bueno [2]

1   Department of Design and Manufacturing Engineering, University of Zaragoza, 50009 Zaragoza, Spain
2   Education Department, Universidad Villanueva, 28034 Madrid, Spain; helena.lopez@villanueva.edu
*   Correspondence: sonia@unizar.es

**Abstract:** The quality of teacher education plays a key role in equipping students with the skills they need and it is important in avoiding learning inequalities. To bridge the digital divide and ensure that all students have equal access to technology and digital resources as well as advanced teaching–learning processes using digital tools, it is crucial to analyse the current state of teacher education in order to identify the key issues. The sample in this research consisted of 325 master of education students and in-service teachers studying at various universities (in Spain, Cyprus, and México) in the 2022–2023 academic year who filled out a questionnaire partly based on DigComEdu and this was expanded with questions relating to other digital and educational aspects. The findings reveal that there is potential for enhancement in many areas. Despite teachers having technical training associated with their profession, they lack the necessary training to overcome inequalities or digital gaps. Moreover, it is striking that those who are already working as teachers are the most pessimistic about aspects such as the use of digital resources, perhaps because of the difficulties they face in their daily work.

**Keywords:** digital divide; digital skills; teacher training



## 1. Introduction

The widespread presence of digital technology in everyday life has substantially transformed the habits of today's society, changing everything from the way we communicate to the way we obtain information and, of course, modifying the ways in which teaching occurs.

The younger segments of society are growing up in a world where digital technologies are ubiquitous, but this does not mean that they have the skills to use these technologies appropriately and wisely [1,2].

Information, technology, and knowledge are clearly interconnected, and this has a clear impact on the way students learn and the way teachers teach [3]. Until recently, learners were mere recipients of content, whereas today they are consumers and generators of knowledge and can consume this knowledge autonomously depending on the context and circumstances, such as during the COVID-19 pandemic [4,5].

However, despite the fact that these students are supposedly digital natives, when it comes to acting effectively and responsibly with digital technology, it is observed that they have serious deficiencies; many do not know how to access information or what criteria to use to discern what is useful and what is not, how to evaluate what they consume and produce, or how to protect their privacy and security, as well as that of the collective [6]. Therefore, it is necessary to effectively integrate ICT in schools and classrooms in order to make it part of quality lifelong learning, allowing students to have access to information and knowledge and to be able to participate fully in society, i.e., to be able to develop the necessary skills that will enable them to integrate digital skills into their lives and work [7].

The key to this transformation of pedagogy in this century and the subsequent empowerment of learners is none other than teachers and their teaching competencies, which

play a fundamental role in guiding learners in the acquisition of ICT skills and abilities [8]. Teacher training and specific and continuous professional development are essential to fully exploit the potential of digital technologies [9], and thus improve teaching and learning for new generations and prepare them for the digital society in which they are immersed [10,11]. Nevertheless, in practice, there is a gap due to digital competency itself, i.e., there is a lack of knowledge about how to make safe and critical use of digital technologies, which directly involves teachers and their own competencies, as transmitters of knowledge and methodologies. Teachers must be able to guide their students in the appropriate use of ICT and help them understand the risks they are exposed to [12]. Moreover, they can contribute to reducing the digital divide by trying to overcome or reduce socioeconomic barriers as much as possible.

The digital divide is a complex concept that manifests in different areas: on the one hand, we have a gap in access to the internet and devices, since a large percentage of the world's population do not have access to the internet, fundamentally due to economic issues, which is a major reason for "technological exclusion", and on the other hand, there is a gap due to a lack of digital knowledge itself.

For this reason, there is a growing need and interest in equipping teachers with digital competencies by creating diagnostic frameworks and tools that help them to self-assess in order to receive training and guidance programs based on their needs in this area.

In Europe, the Digital Competence Framework for Citizens (DigComp) was first published in 2013 [13]. It was the prelude to the so-called European Framework for Digital Competence of Educators, DigCompEdu [14], which aimed to group the elements of an educator's digital competence into a number of main areas and to establish indicators of progression in each of these areas.

Subsequently, to achieve this digital competence, the European Commission developed the Digital Education Action Plan (2021–2027) [15] to support the sustainable and effective adaptation of education and training systems in EU Member States to the digital age. It has two priorities: The first is to foster the development of a high-performing digital education ecosystem, including digitally competent and secure teachers and educational and training staff, high-quality learning content, user-friendly tools, and secure platforms that respect e-privacy and ethical standards and effectively address digital divides. The second priority is to improve digital skills and competencies for digital transformation by establishing common guidelines for teachers and educators to promote digital literacy and address disinformation through education and training.

Other initiatives also explore aspects of digitalization for teachers. For example, JISC [16] investigated how educational skills can be improved through the use of digital media and the impact that generative artificial intelligence is having on education or the Global Framework for Educational Competence in the Digital Age, an international competency framework for educators.

Teachers' digital skills can be passed on to students [17], which will improve their digital competencies and justify the use of digital media, as well as eliminating gaps of all kinds associated with digitalization. Consequently, teachers will be among the main actors, as they are responsible for transmitting knowledge, so digital training for teaching in all aspects should be basic and must be taken into account in training programs. So far, the data indicate that in general, training in digital skills for teachers is almost entirely based on self-learning, with the use of hardware and software or by viewing courses through social networks [18].

In the teaching field, digital competence and digital technologies have been integrated to carry out various activities, such as presentations and specific tasks through the use of specialized software. However, it is important to note that these tools are mainly used for pedagogical support and are not fundamental components in the essential training of teachers [19,20].

However, due to the exceptional circumstances during the COVID-19 pandemic, these technologies were urgently implemented in an attempt to continue with the rhythm

of classes, such as for handing in homework and carrying out tutorials. At this time, difficulties in using technology for both students and teachers were revealed [21].

In this sense [10], the results of the questionnaire in the present research indicate that only a small percentage of the surveyed teachers (approximately one-third) were capable of mastering the digital environment and associated tasks. Consequently, they did not act as catalysts for learning in the digital setting. This underscores the need to revisit the curricula of teacher training courses and postgraduate programs, for both those in undergraduate programs and those in training who may also be active teachers [22,23]. To do this, it is necessary to know the level of digital competency of teachers or teacher trainees [24–26] in order to implement effective strategies to improve their digital competence in those areas where it is most needed, and provide them with tools not only to generate digital content but also to address digital gaps in students' abilities and the risks involved. Going deeper into the subject, as concluded in [27], the existence of technological resources alone is not a sufficient condition for competencies to be well addressed.

Considering the importance of the digital aspects of continuing education and training of teachers, it is necessary to assess the digital education content of master's programs in education and the possible shortcomings perceived by students in these programs (who may be future teachers or are already in-service teachers). Thus, there was a need for this study, which analyses the digital competencies of master of education students and in-service teachers studying at several universities in the 2022–2023 academic year, namely: University of Zaragoza and Autonomous University of Madrid, Spain, University of Nicosia, Cyprus, and Popular Autonomous University of the State of Puebla and National Pedagogical University, Mexico. The aim was to see the situation of future teachers in terms of digital skills and the possibility of bridging any digital gaps, so that we can assess strengths and possible areas for improvement.

## 2. Materials and Methods

Considering the purpose of this study, a quantitative non-experimental model with an ex post facto design was chosen. Specifically, the descriptive method used was the survey method, with the formulation of direct questions from a previously prepared scripted protocol presented to a representative sample of subjects.

The information was collected using two identical Google Forms questionnaires, in English and Spanish, administered in the 2022–2023 academic year to master of education students at the above-mentioned universities. Anonymity was guaranteed for participation.

### 2.1. Sample

The sample was composed of master's degree in education students and in-service professors studying at several universities in the 2022–2023 academic year: University of Zaragoza and Autonomous University of Madrid, Spain, University of Nicosia, Cyprus, and Popular Autonomous University of the State of Puebla, Mexico. All individuals were assigned the same probability of selection based on chance, i.e., the criterion was one of equiprobability. The sample of participants consisted of 325 postgraduate students whose training enabled them to be teachers at different educational stages.

Most of the respondents were in the 30–39 age group, and the majority, 73.5%, were women, compared to 26.2% men and 0.3% identifying with other forms of gender expression. As for the area of origin, 74.77% were from urban areas or nuclei, 14.77% from semi-urban environments, and 10.46% from rural areas.

### 2.2. Instrument

The instrument was a questionnaire partially based on the DigCompEdu survey, complemented with questions about social justice and the digital divide and other questions (https://docs.google.com/forms/d/1s_TqgYozoyXyJmW11jd3tT4J1WK_wQu11opBJmgmUgM/prefill, accessed on 15 March 2024). The instrument was validated by six educational experts from various universities (two experts in Spain, one expert each in México, France, and Belgium).

All participants were given prior information about the motivation for the survey, and their anonymity was ensured. A Likert scale was used for questions for which there were scaled responses: 1 = strongly disagree to 5 = strongly agree.

### 2.3. Data Analysis

The data analysis was performed using IBM SPSS Statistics v24 software to assess the consistency of the questionnaire with the calculation of Cronbach's alpha, which was 9.962, which indicates a high level of reliability [28].

### 3. Results

Our main objective was to assess the digital competence of teachers and to see how those who are training to be teachers or already are teachers and want to improve their skills perceive issues such as social justice and the digital divide, which are fundamental for their professional performance in relation to their training or self-perception.

The following sections present the main results obtained from the sample.

### 3.1. Relationship with Technology and the Potential of Technology to Improve Educational Processes

The part of the questionnaire asking about digital competence started with a self-assessment question about the respondent's relationship with technology.

The results (Table 1) show that the majority of respondents rated themselves as normal or conventional (68.6%) and the rest as innovative or advanced (31.4%).

**Table 1.** Respondents' relationship with technology.

| Relationship with Technology | (%) |
|---|---|
| Advanced | 11.7 |
| Innovative | 19.7 |
| Conventional | 3.4 |
| Normal | 65.2 |

It might be expected that "digital natives", or at least younger respondents, would consider themselves to have an innovative or advanced relationship with technology, but this was not the case (Table 2). Those in the 30 to 39 year age group considered themselves to be the most advanced in their self-assessment. In terms of gender, the self-assessment was very similar in all categories, except for advanced, which had a higher percentage of men.

**Table 2.** Self-assessment of relationship with technology by age group.

| Relationship with Technology (%) | Age (Years) | | | | |
|---|---|---|---|---|---|
| | <25 | 25–29 | 30–39 | 40–49 | 50–60 |
| Advanced | 1.2 | 3.1 | 5.5 | 1.2 | 0.6 |
| Innovative | 3.1 | 6.2 | 6.8 | 2.8 | 0.9 |
| Conventional | 0.3 | 0.6 | 1.8 | 0.6 | 0.0 |
| Normal | 11.7 | 12.9 | 26.5 | 9.2 | 4.9 |

Obviously, the relationship that teachers have with technology has a direct bearing on whether or not they will use it in their teaching process and contribute to its improvement. For this reason, a question was included regarding the respondents' perception of whether or not technology could improve the educational process.

The following data were obtained from cross-referencing the questions "What is your relationship with technology?" and "According to your point of view, can technology improve the quality of educational processes in all cases/situations?" (Table 3).

**Table 3.** Cross-reference between self-assessment of relationship with technology and opinion on whether technology can improve educational processes.

| Relationship with Technology | Technology Can Improve Educational Processes (%) | | | | |
|---|---|---|---|---|---|
| | 1 | 2 | 3 | 4 | 5 |
| Advanced | 0 | 2.6 | 10.5 | 42.1 | 44.8 |
| Innovative | 0 | 0 | 9.4 | 39.1 | 51.5 |
| Conventional | 0 | 9.1 | 18.2 | 45.5 | 27.2 |
| Normal | 1.4 | 3.3 | 16.5 | 48.1 | 30.7 |

We observed that, in general, most of the respondents thought that technology can improve educational processes, regardless of their digital background. In all cases, the greater the self-perception of digital competence, the greater the belief that technology can improve the quality of educational processes.

In the same way, we might think that younger respondents would be more likely to think that the use of technology can improve the quality of educational processes, but this was not the case; the majority of respondents who thought this were in the 30 to 39 year age group (Table 4).

**Table 4.** Relationship between belief that technology can improve educational processes and age of respondents.

| Age | Technology Can Improve Educational Processes (%) | | | | |
|---|---|---|---|---|---|
| | 1 | 2 | 3 | 4 | 5 |
| <25 | 5.70 | 7.50 | 39.60 | 26.40 | 20.80 |
| 25–29 | 0.00 | 12.20 | 33.80 | 33.70 | 20.30 |
| 30–39 | 1.50 | 12.10 | 20.50 | 43.90 | 22.00 |
| 40–49 | 0.00 | 8.90 | 28.90 | 37.80 | 24.40 |
| 50–59 | 0.00 | 0.00 | 52.40 | 19.00 | 28.60 |

Next, a cross-check was made between the answers related to the respondents' current occupation and whether the use of technology can improve learning processes, and the results are shown in Table 5.

**Table 5.** Relationship between opinion on whether technology can improve educational processes and respondent's current occupation.

| Current Occupation | Technology Can Improve Educational Processes (%) | | | | |
|---|---|---|---|---|---|
| | 1 | 2 | 3 | 4 | 5 |
| Master's student | 0.31 | 1.54 | 6.77 | 18.46 | 12.92 |
| Master's student and other work | 0.62 | 0.92 | 4.31 | 16.62 | 17.85 |
| Master's student and teacher | 0.00 | 0.31 | 3.38 | 10.46 | 5.54 |

Table 5 shows that the most optimistic responses were from students who were studying for a master's degree and worked in something unrelated to teaching, with 34.47% indicating "agree" or "strongly agree". This is followed by those whose only occupation was their master's degree studies, among whom 31.38% indicated "agree" or "strongly agree". Finally, the most pessimistic were those who were working as professors and studying for a master's degree, with 16% for these two categories.

To delve into the issue, we looked at the respondents' opinions regarding whether computers, tablets, and other digital devices are used in schools as a basis for teaching and learning (Figure 1).

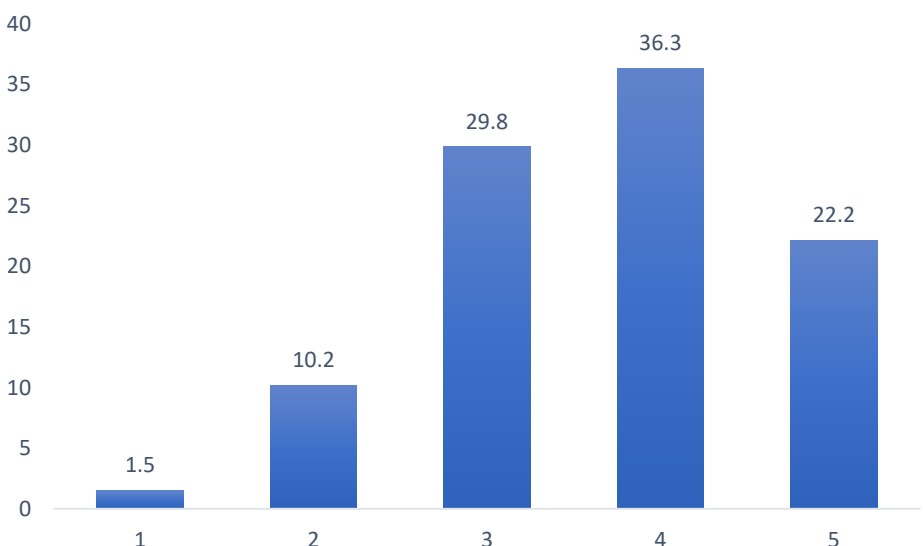

**Figure 1.** Assessment of whether computers, tablets, and other digital devices are used in schools as a basis for the teaching–learning process (%).

Almost 42% of the respondents indicated that they believed digital devices are not primarily being used as a basis for teaching and learning (sum of scores 1, 2, and 3). That means that teachers in service do not see digital devices as strategic tools for educational development. However, if we cross-reference this response with occupation, the results are surprising (Table 6).

**Table 6.** Relationship between respondents' current occupation and their assessment of computers, tablets, and other digital devices being used as a basis for the teaching–learning process.

| Current Occupation | Assessment of Computers, Tablets, and Other Digital Devices Being Used in Schools as a Basis for the Teaching–Learning Process (%) | | | | |
| --- | --- | --- | --- | --- | --- |
| | **1** | **2** | **3** | **4** | **5** |
| Master's student | 0.31 | 4.92 | 13.85 | 13.85 | 7.08 |
| Master's student and other work | 0.92 | 3.69 | 8.92 | 16.00 | 10.77 |
| Master's student and teacher | 0.31 | 1.54 | 7.08 | 6.46 | 4.31 |

The data suggest that those who are studying for a master's degree or working in a non-teaching job while studying for a master's degree are more optimistic about the use of digital devices. This is somewhat worrying, as it suggests that those with teaching experience may not be sufficiently prepared, or their experience may have made them aware of the difficulties in accessing hardware and internet connectivity.

### 3.2. Assessing the Potential for Addressing the Digital Divide in Education

The results reported in the previous section are important to know the respondents' digital level and general assessments of educational processes, especially those that include the use of digital resources. It is also crucial to know whether the respondents have the necessary training to address the gaps in the digitalization of education, in terms of inequalities in access and training. These data, cross-referenced with variables such as occupation and the use of open resources, give us information about the current situation of teachers who are training or students studying to be teachers.

The relationship between the training needed to address the digital divide and respondents' current occupation is shown in Table 7.

**Table 7.** Relationship between training needed to address digital divide and respondents' current occupation.

| Current Occupation | I Have the Necessary Training to Address the Gaps When It Comes to the Digitalization of Education (%) | | | | |
|---|---|---|---|---|---|
| | **1** | **2** | **3** | **4** | **5** |
| Master's student | 2.15 | 5.85 | 13.54 | 13.54 | 4.92 |
| Master's student and other work | 1.54 | 4.31 | 11.69 | 16.31 | 6.46 |
| Master's student and teacher | 0.62 | 1.54 | 9.54 | 6.77 | 1.23 |

In absolute terms, 34.77% of respondents said "slightly agree" and 36.62% said "somewhat agree". These data are encouraging but not optimal, as they show that only 12.62% of respondents strongly agree, while 16% strongly disagree or somewhat disagree (sum of levels 1 and 2). It would be desirable for practically all teaching staff and levels to be between the values in columns 4 and 5, especially if they are training in a postgraduate course related to teaching, in order to be able to think that the digital divide is being addressed adequately and effectively.

If, at the same time, we analyze the relationship that respondents believe exists between teachers' use of technology and the transformation of learning processes, the results are, in some way, a complement to the previous answer (Figure 2).

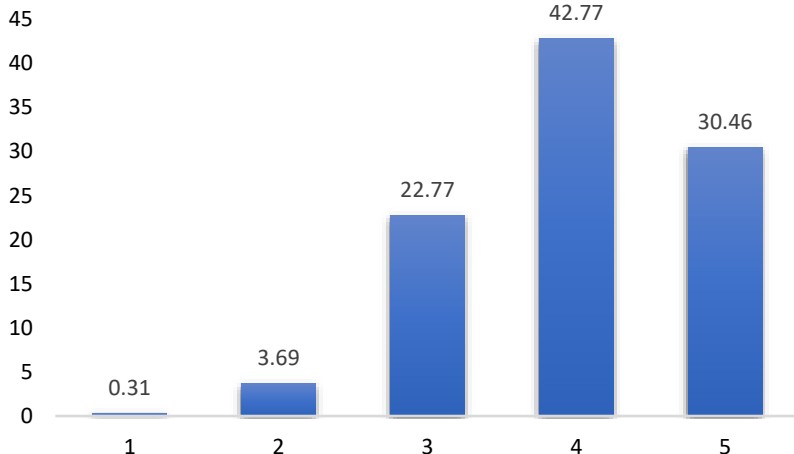

**Figure 2.** Assessment of whether there is a direct relationship between the use of technology and the transformation of learning processes.

Mostly, the respondents agreed that the relationship is direct and that the use of technology can transform learning processes (the sum of responses rated 4 and 5 is 73.23%).

When we cross-reference responses to the statement "My educational center (university) does not provide me with sufficient training to use ICT that may serve in the future to offer inclusive education and guarantee real equal opportunities in my activities as a future teacher" with the responses to "Relationship with technology", we obtain the data shown in Table 8.

The data show that the more advanced the respondent's relationship with technology, the less they agree with the statement. If we analyse category by category, we find the following:

- Advanced: The highest percentage of respondents (42.10%) rated their ICT training as sufficient (score of 3) in offering an inclusive education and ensuring equal opportunities. However, a significant portion rated it as insufficient (13.20%) or almost insufficient (26.30%).
- Innovative: A notable percentage of respondents (36.40%) rated their training as somewhat sufficient, but a substantial number (27.30%) considered it to be insufficient.

- Conventional and normal: The answers are more evenly distributed between scores of 1 to 5, with more considering the training as insufficient or almost insufficient than those considering it sufficient or strongly sufficient.

**Table 8.** Relationship between responses to "My educational center (university) does not provide me with sufficient training to use ICT that may serve in the future to offer inclusive education and guarantee real equal opportunities in my activities as a future teacher" with "Relationship with technology".

| Relationship with Technology | My Educational Center (University) Does Not Provide Me with Sufficient Training to Use ICT That May Serve in the Future to Offer Inclusive Education and Guarantee Real Equal Opportunities in My Activities as a Future Teacher (%) | | | | |
|---|---|---|---|---|---|
| | 1 | 2 | 3 | 4 | 5 |
| Advanced | 13.20 | 26.30 | 42.10 | 7.90 | 10.50 |
| Innovative | 18.20 | 36.40 | 27.30 | 9.10 | 9.10 |
| Conventional | 23.40 | 17.20 | 25.00 | 17.20 | 17.20 |
| Normal | 17.90 | 22.20 | 29.70 | 18.90 | 11.30 |
| Total | 18.50 | 22.20 | 30.20 | 16.90 | 12.30 |

Overall, the table suggests that while a significant portion of respondents perceive their ICT training to be sufficient (especially in the advanced category), a notable percentage rate it as insufficient, particularly in the innovative and conventional categories. This indicates a need for improvement in digital training to adequately prepare teachers to offer inclusive education and ensure equal opportunities.

On the other hand, when we look at the responses to "I believe that I have the necessary training to address the gaps regarding the digitization of education (inequality in access, inequality in training, and inequality in support)", we obtain the data shown in Figure 3.

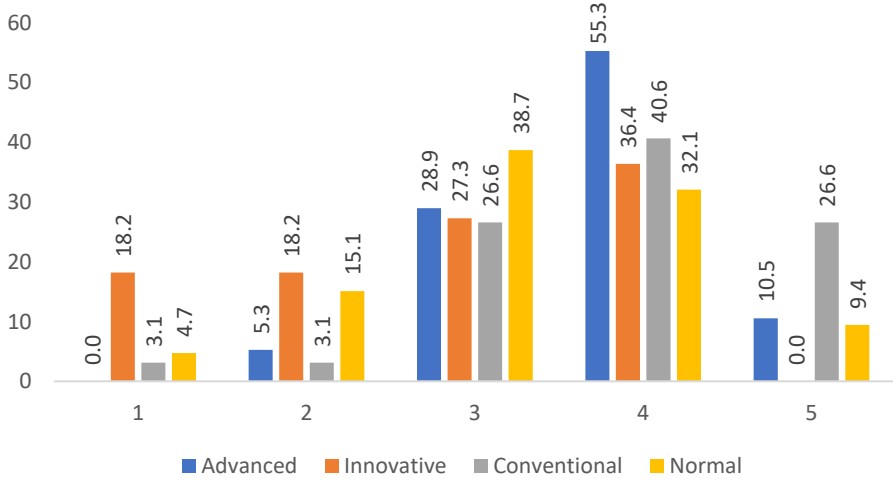

**Figure 3.** Responses to "I believe that I have the necessary training to address the gaps regarding the digitalization of education (inequality in access, inequality in training, and inequality in support)".

Those in the advanced category indicated that they feel well-prepared to address the gaps related to digitalization in education, with the highest percentage (55.3%) giving a rating of 4, indicating strong confidence in their training. However, a notable portion of this category (28.9%) gave a rating of 3, indicating some confidence but with room for improvement.

The responses of those in the innovative category varied, with a significant percentage (36.4%) giving a rating of 4, indicating confidence, but a substantial portion (27.3%) indicating a moderate level of confidence.

The responses of those in the conventional category show a similar trend to the innovative group, with a significant portion (40.6%) giving a rating of 4, and aligning with the innovative group, a higher percentage (26.6%) gave a rating of 2, indicating less confidence in their preparedness.

The responses in the normal category are distributed between 1 and 5, with most (38.7%) giving a rating of 3, indicating moderate confidence, and the rest of answers indicating mixed feelings about their perceived preparedness.

It can be said that, except for the advanced group, there is a wide range of levels of confidence in their training. This suggests that there may be a need for further training and support to ensure that all educators feel equipped to tackle inequalities in access, training, and support in the context of digital education [29].

### 3.3. Training Needs

Training is a key factor in improving the performance of teachers and providing them with strategies to not only address the digital divide but also other issues related to the teaching–learning process apart from the transmission of knowledge, such as training in soft skills.

Thus, educational centers should incorporate specific sections in their training programs that allow students to acquire these skills. The respondents were asked about the training they received in their postgraduate courses on using ICT in order to offer an inclusive education that guarantees real equality of opportunities in their activities as future teachers, and their answers are summed up in Figure 4.

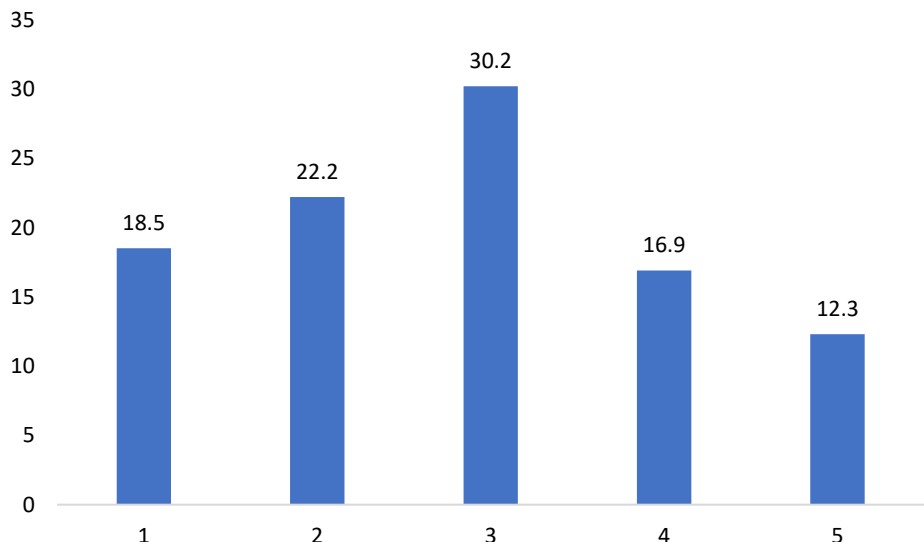

**Figure 4.** Responses to "My school (university) provides me with sufficient training to use ICT that can be used in the future to offer inclusive education".

Grouping the answers into blocks, we can see that 30.20% of respondents "slightly agree", 40.70% "totally agree" or "partially disagree", and 29.20% "totally agree" or "partially agree", so it seems that only those in the latter block (a minority) were satisfied with the training they were receiving. Most responses were rated as 3, indicating the perception that the ICT training is moderately sufficient. Despite the different perceptions of respondents, some results can be highlighted from the data related to the different categories.

The majority of respondents who considered themselves to be advanced (42.10%) rated their training as 3, indicating that they believe it to be sufficient. However, a notable percentage also rated their training as 1 or 2, indicating some inadequacy, although this category accounts for 38 out of 325 responses (11.69% of the total).

Among those who considered themselves to be conventional, the highest percentage (36.40%) gave a rating of 2, suggesting a perception of inadequacy in ICT training. This category accounts for 11 out of 325 responses (3.38% of the total).

Among those who considered themselves to be innovative, most of the answers (25.00%) were rated 3, indicating perceived sufficiency in training. This category accounts for 64 out of 325 responses (19.69% of the total).

The answers for the normal category are similar to those for innovative, 29.70% giving a rating of 3, but this category accounts for the highest number of responses, 212 out of 325 (65.23% of the total).

The respondents were also asked about the areas in which they believe they need more training, and the answers, in descending order, are shown in Figure 5.

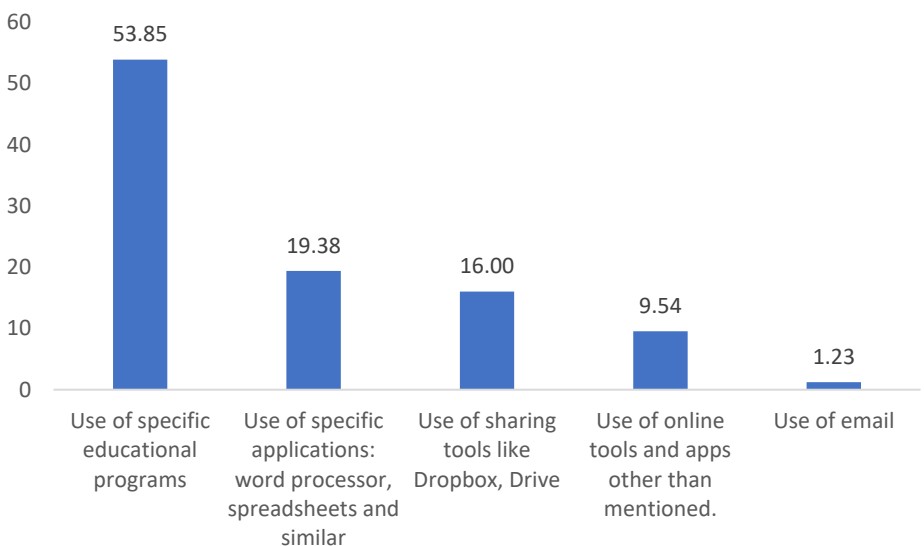

**Figure 5.** Respondents' views of areas in which they need more training.

As can be seen, the category in which the most students demand more training is specific educational programs, accounting for more than half of the responses. There are many programs for different areas and educational objectives, but on many occasions, beyond their complexity, the problem is that they are unknown to teachers.

Other interesting indicators include those related to the use of open resources, the results of which are shown in Table 9.

**Table 9.** Ability to use open educational resources and licenses.

| Variable | (%) | | | | | |
|---|---|---|---|---|---|---|
| | **1** | **2** | **3** | **4** | **5** | **4 + 5** |
| I know how to use open educational resources | 2.77 | 10.46 | 33.85 | 31.38 | 21.54 | 52.92 |
| I know how to correctly use licenses (use and creation) of open educational resources, including correct attribution | 7.69 | 22.46 | 30.77 | 23.08 | 16.00 | 39.08 |
| I know how to modify and adapt existing open licensed resources, as well as other resources where this is allowed | 13.23 | 18.15 | 31.69 | 22.15 | 14.77 | 36.92 |

It would be desirable for most of the respondents' answers to be in the last two columns, but when adding the values in columns 4 and 5, the first question alone accounts for just over half of the answers. The answers to the other two questions indicate a general lack of knowledge about the use and modification of licenses for open educational resources. These resources allow free access to many utilities and resources that could help reduce the

digital divide in some cases, and if teachers do not know how to use them, they will not be used.

The data show that the majority of respondents have at least a moderate level of knowledge about using open educational resources and modifying and adapting existing resources. With respect to the use of licenses, the answers indicate a higher level of knowledge in this aspect compared to the previous one.

Other questions were aimed at finding out the degree of knowledge and autonomy of students, and the data obtained are given in Table 10.

**Table 10.** Respondents' self-reported levels of knowledge regarding various important issues related to digital education.

| Variable | (%) | | | | |
|---|---|---|---|---|---|
| | 1 | 2 | 3 | 4 | 5 |
| I know how to create, individually or in collaboration with others, new digital educational resources | 8.0 | 20.0 | 32.6 | 23.1 | 16.3 |
| I know how to protect confidential information | 8.6 | 16.9 | 22.5 | 29.8 | 22.2 |
| I know how to correctly apply the regulations on privacy and intellectual property | 9.2 | 18.2 | 28.0 | 24.6 | 20.0 |

In response to the first question, the majority of respondents rated their knowledge in this area as moderate to high. Specifically, 32.6% gave a rating of 3, and 23.1% a rating of 4. This means that they feel self-confident about this task although the level of help required is unknown. Similar results were found for the second question, with 29.8% rating their knowledge as 4, and 22.5% as 3. For the third question, most respondents rated their knowledge in this area as moderate to high, but a significant portion (20.0%) rated their knowledge as 5, indicating a high level of knowledge.

In general, respondents indicated a moderate to high level of knowledge regarding important issues in digital education, such as creating digital educational resources, protecting confidential information, and applying regulations on privacy and intellectual property. However, there is still room for improvement, particularly in areas where fewer respondents rated their knowledge at higher levels. Additional training or resources may be beneficial in further enhancing people's understanding and proficiency in these areas.

## 4. Discussion

From the results, interesting data were obtained related to the digital competence of in-service teachers who are being trained or those whose training is preparing them for the practice of the profession. Part of this digital competence is related not only to their own training but also to their ability to improve educational processes through the use of digital technologies and overcome barriers such as the digital divide [30,31], which can put some of their students at risk of exclusion.

Regarding the relationship with digital technologies, the majority of respondents still maintain a normal or conventional profile (68.6%). This indicates that there is a great deal of room for improvement, which could be achieved in undergraduate and postgraduate teacher training. For digital skills to be well managed, material resources will be essential, but also increasing training in digital skills for teachers will improve the quality of education related to these technologies. This finding is aligned with ElSayary [32] who found that teachers gap of competences has a direct influence in students learning and in the innovative use of digital technology.

However, from the results obtained, it can be seen that most of the participants believe that technology can help to improve educational processes, especially those who, although they are pursuing a master's degree related to education, work in a field other than teaching [33]. By age, it is mainly the 30-to-39-year-old group who believe this, perhaps because they have had the opportunity to compare the current possibilities for

the use of ICT with those at the beginning of their professional career and they value the improvement [34].

It can be observed from the results that in terms of the respondents' self-perception of their relationship with technology, women have a much lower perception than men, and we could say that they are more skeptical that technology is capable of improving teaching–learning processes. This is a gap that would need to be reviewed in depth to analyze the causes. According to [35], women need to know a topic, that is, master it, in order to be confident, so their answers are usually more conservative. This should be taken into account to compensate for inequalities in the education of women and girls [36].

After looking at digital skills, we analyzed whether they are sufficient to address other digital divides. The results indicate that although teachers do have technical training related to their professional performance, they lack the necessary training to overcome the inequalities that exist as a result of the different types of digital divide. It is observed that even though the majority believe that they have the knowledge to overcome the digital divide, only 12.62% of the responses indicate total agreement. In this sense, it would be desirable that all respondents gave a rating of 4 or 5, as it would be a signal of success in overcoming the digital divide at all levels.

If we look specifically at the current occupation of the respondents, we again observe that those who already work as teachers are the most pessimistic, perhaps because they are aware of the difficulties they face in their day-to-day lives. Most believe that it is possible to transform learning processes by using technology, but without knowing how to address the inequalities that can arise (economic, access to resources depending on the geographical area, etc.), it seems complicated to make a real and consistent transformation [37]. While it is true that those who consider themselves to be advanced/innovative are seen as more capable of bridging the digital divide, their numbers are still small. Numerous authors have addressed this aspect and have reached a similar conclusion, i.e., that future teachers have insufficient digital competence to address certain problems of this type [8,38,39].

Therefore, teacher training will be a determining factor in overcoming digital barriers of all kinds, as well as improving the teaching–learning process. It is essential that universities value the importance of this training because of the repercussions it will have at a social and economic level on addressing specific curricular actions, not only in the medium and long term, but also in the short term [40]. In addition, education administrators should provide lifelong learning programs for teachers to learn, improve, or complement digital skills, in which training is expanded in specific aspects such as choosing methodologies, reducing inequalities, or overcoming the digital divide.

## 5. Conclusions

In the context of rapid evolution towards an increasingly digitized society, the significance of digital competencies for teachers is magnified in terms of future readiness [41]. The insightful data obtained in research on the digital competence of in-service teachers not only reflects the current situation but also underscores the urgent need to cultivate robust digital competencies to address the challenges of tomorrow [42]. As technology continues to transform education and the workforce, educators must acquire advanced digital skills that go beyond mere interaction with technological tools.

In this sense, the fact that the majority of respondents consider themselves to have a normal or conventional relationship with technology highlights a significant opportunity to develop more advanced digital skills. The gap between current skills and those required to tackle future complexities, such as using artificial intelligence in education [43], adapting to emerging technological trends, and addressing problems associated with equity in technology access, indicates an immediate need for a more comprehensive educational approach.

Increasing digitalization also implies greater global interconnectedness, which highlights the importance of intercultural skills and the ability to work together in virtual environments. Teacher training should go beyond the acquisition of technical skills [44] and address the ability to navigate in a diverse and globalised digital world. In addition, aware-

ness of digital ethics and online safety is crucial to ensure responsible use of technology and prepare educators to guide their students in understanding these fundamental issues.

Finally, teacher education should focus on the development of advanced and adaptable digital literacy that will enable teachers not only to take advantage of the opportunities offered by technology, but also to address the ethical, social, and economic challenges that may arise in the future [45]. Integrating these competencies into initial and continuing education programs is essential to prepare educators as change agents capable of leading the digital transformation of education and contributing to the development of more inclusive and resilient societies.

**Author Contributions:** Conceptualization, S.V. and H.L.-B.; methodology, S.V. and H.L.-B.; validation, S.V. and H.L.-B.; formal analysis, S.V.; investigation, S.V. and H.L.-B.; data curation, S.V.; writing—original draft preparation, S.V.; writing—review and editing, S.V. and H.L.-B. All authors have read and agreed to the published version of the manuscript.

**Funding:** This research received no external funding.

**Institutional Review Board Statement:** Not applicable.

**Informed Consent Statement:** Not applicable.

**Data Availability Statement:** Any inquiries can be directed to the corresponding author.

**Conflicts of Interest:** The authors declare no conflicts of interest.

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
