# Peer review of "Analysis of Digital Teacher Education: Key Aspects for Bridging the Digital Divide and Improving the Teaching–Learning Process"

_education, doi:10.3390/educsci14030321_

Round 1

Reviewer 1 Report

Comments and Suggestions for Authors

Title

The title is appropriate.

Abstract

The abstract can benefit from revision at the moment the problem is not very clear. The statement on the results can be more specific rathe than just “results show that there is room for improvement’

Introduction

The introduction is adequate but the justification as to why this study is important in teaching and learning.

Methods

The selection or recruitment strategies need to be explained. Were the different authors involved in collecting the data from their institution. The sampling needs to be revised so that the wording is consistent, for example on line 122, the word professors is used.

Results

This is the section that will benefit from revision. The tables need to be self-explanatory. I suggest that instead of using 1 to 5, the actual Likert words should be used. I also noticed that the Figures have lo labels. It is important that the reporting of these results focus on key findings rather than trying to explain what is contained in the table.

Sine the sample size is big is there a reason why inferential statistics was not used.

There is a need to check if the sentences are written n full. For example, line 284, maybe start this this sentence using a group of participants.  The results presented are adequate and it will assist if these results are presented using the research questions.

Discussion

The discussion can be strengthened by citing more studies and an attempt to theories the finding can be helpful.

The conclusions should provide answers to the research questions.

Author Response

Dear reviewer

Thank you for your review. Your comments have been included and help us to improve the content. Below, a point-by-point response:

  • Title has been changed.
  • Abstract was modified following all the directions.
  • In sections Introduction, Methods all the comments have been addressed.
  • References have been extended for wide literature review.
  • Conclusions sections has been created.
  • English editing has been done by MDPI english services.

Additionally, we used MDPI english edition services for this english version.

Reviewer 2 Report

Comments and Suggestions for Authors

Dear Authors, 

Please find my main comments in the attached document.

Kind regards.

Comments on the Quality of English Language

Check the entire document for English language mistakes (e.g. Lines 13-14, 35-37, and others do not make sense).

Author Response

Dear reviewer

Thank you for your review. Your comments have been included and help us to improve the content. Below, a point-by-point response:

  • Reference to different digital frameworks has been addressed in Introduction section.
  • Additionally, other changes have been done in all sections for improving quality.
  • English editing has been done by MDPI english services.

Reviewer 3 Report

Comments and Suggestions for Authors

Dear Author(s), 

Thank you for giving me the opportunity to review your interesting study. The quality of the study is strong. However, I would like to see some more engagement with the relevant digital literacy/competency frameworks in your introduction section. For example, I expected to see reference to frameworks such as DigiComp 2.2, DigiCompEdu, JISC, among others. I think including a review of these frameworks and their implications would significantly improve the study. 

Author Response

Dear reviewer

Thank you for your revision. We upload the manuscript with all changes adressed and English Editing by MDPI English Editing services.

Round 2

Reviewer 2 Report

Comments and Suggestions for Authors

Dear authors,

I see you have integrated most of my initial comments (for instance, you created a new section, Conclusions, which wasn't present in the first version of the paper). I still feel, however, that you could improve the Discussions by linking your comments (which are pertinent) to the findings in other studies. Are your results in agreement with previous works? This could be clarified. Something similar to what you did in Line 432, where you brought 3 references to support your idea.

Author Response

Dear reviewer

Section Discussion has been improved by adding references to other researches whose findings are consistent with ours. We have selected this references  in our draft version. Changes have been highlighted in green.

Thank you. Regards

Reviewer 3 Report

Comments and Suggestions for Authors

Dear Author(s), 

Thank you for your revised paper and for the work you carried out during this process. 

Author Response

Dear reviewer

In addition, Discussion section has been improved by adding references to other research whose findings are consistent with ours. We have selected these references in our draft version. The changes have been highlighted in green.

Thank you very much
